# LRN: Limitless Routing Networks for Effective Multi-task Learning

## Abstract

Multi-task learning (MTL) aims at learning multiple tasks simultaneously typically through shared model parameters. The shared representation enables generalized parameters that are task invariant and assists in learning tasks with sparse data. However, the presence of unforeseen task interference can cause one task to improve at the detriment of another. A recent paradigm constructed to tackle these types of problems is the routing network, which builds neural network architectures from a set of modules conditioned on the input instance, task, and previous output of other modules. This approach has many constraints, so we propose the Limitless Routing Network (LRN) which removes the constraints through the usage of a transformer-based router and a reevaluation of the state and action space. We also provide a simple solution to the module collapse problem and display superior accuracy performance over several MTL benchmarks compared to the original routing network.

## 1 Introduction

Multi-task learning (Caruana, 1997) is the objective of learning multiple tasks simultaneously with some shared representation, such as shared parameters in a deep neural network. The shared parameters can enable knowledge gained from one task to be leveraged for successfully completing another, which is called positive transfer. However, this is not always the case. As model parameters are updated, not only can information be lost through catastrophic forgetting (McCloskey & Cohen, 1989; Ratcliff, 1990; Kirkpatrick et al., 2017), but can move in a direction that could negatively affect the performance of another task, called negative transfer.

In (Rosenbaum et al., 2017), researchers propose the routing network which is comprised of a router and a set of modules, where the modules in their experiments consist of fully-connected (FC) layers. The router constructs a subset of a neural network's layers by recursively routing the output of a previously selected module through another, conditioned on the input instance, task, and output of the previous module until a particular depth is reached. This allows for conditionally using different model parameters based on the input in hopes it can reduce negative transfer and catastrophic forgetting while preserving positive transfer. However, this approach has several constraints: 1) It requires the module output size to be fixed as the router requires the output of a module as part of the state; 2) Each module belongs strictly to a particular depth; and 3) The total number of modules at every level of depth is fixed.

As these limitations enforce constraints on the feasible region of modular architecture, we propose the *Limitless Routing Network (LRN)* which removes the aforementioned constraints through the usage of a transformer-based router and a reevaluation of the state and action space. We also extend routing to every layer in the constructed network and increase the diversity of modules to show the capabilities and generalizability of our approach. In addition to removing the constraints and increasing functionality, we propose a simple solution to help prevent module collapse (Kirsch et al., 2018; Rosenbaum et al., 2019), whereby the router only routes to a subset of modules regardless of the input, through an auxiliary reward.

Through empirical evidence given by testing on all the same MTL benchmarks discussed in (Rosenbaum et al., 2017), we show superior testing accuracy compared to the original routing network. In summary, our contributions are as follows:

- Removing the constraints from the routing network to allow for more complex model construction;
- A novel approach to prevent module collapse and increase model diversity through a simple, auxiliary reward;
- Using function approximation for the router to enable conditioning the constructed network on the instance level information; and
- Allowing routing on every layer of the modular network.

The source code of LRN can be found at `https://anonymous.4open.science/r/TaskRouter-A624`.

## 2 RELATED WORK

Traditionally work on multi-task learning (Caruana, 1997) involves manually creating the shared representation and having it fixed for all tasks. For example, sharing the first few layers of a neural network, but then having different heads for each task. This is similar to what is typically done in Reinforcement Learning and Actor-Critic architectures (Konda & Tsitsiklis, 2000), where the actor and critic have different output heads, one for the policy and the other for the state-value function.

Our work is based on the routing network (Rosenbaum et al., 2017), which uses a multi-agent reinforcement learning (MARL) approach and the weighted policy learner (WPL) algorithm (Abdallah & Lesser, 2006), to jointly train the router and set of modules. All the benchmarks in (Rosenbaum et al., 2017) on are classification tasks, where the reward is set to be +1 if the class is correctly predicted and -1 otherwise. The state space is defined by the tuple $(v, t, i)$, where $v$ is output of the previous module, $t$ is a one-hot encoded vector representing the current task, and $i$ is the current timestep. Their action space is defined over all the modules in the current timestep, which is kept to the same fixed size. In addition, in (Rosenbaum et al., 2017), tabular representation is adopted to represent the policy and found to outperform the function approximation representation. In this work, we show that function approximation does work in our modified approach. A more in-depth analysis of recent work on multi-task learning with deep neural networks can be found in (Crawshaw, 2020).

A similar line of work is that of neural architecture search (NAS) (Zoph & Le, 2017; Baker et al., 2017; Miikkulainen et al., 2017; Liu et al., 2018; Pham et al., 2018), where the goal is to search for neural network architectures that are best fit for a task. This differs from the conditional architecture construction employed in our work as it primarily focuses on finding a singular architecture for all instances rather than having it conditioned on each input instance. An in-depth survey of recent work is discussed in (Ren et al., 2020).

## 3 PRELIMINARIES

For a $T$-step episodic task, an RL agent applies its policy $\pi_\theta(a_t|s_t)$ to choose action $a_t$ in state $s_t$ at each timestep $t \in [1, T]$. Then, the environment responds with reward $r_t$ and next state $s_{t+1}$. This sequential decision-making process is formulated as a Markov Decision Process (MDP) defined by the tuple $(\mathcal{S}, \mathcal{A}, P, R, \gamma)$, where $\mathcal{S}$ is the state space, $\mathcal{A}$ is the action space, $P(s'|s, a)$ is the transition probability function, $R$ is the reward function, and $\gamma \in [0, 1]$ is the discount factor. The objective of the RL agent is to find $\pi_\theta$ that can maximize the sum of discounted rewards $R_t = \sum_{i=t}^{T} \gamma^{i-t} r_i$.

One way to obtain $\pi_\theta$ is via policy gradient methods, which learn $\pi_\theta$ by updating $\theta$ using gradient ascent. $\pi_\theta$ is typically termed *actor*. The policy gradient has the general form (Schulman et al., 2018):

$$\mathbb{E}_{\pi_\theta} \left[ \sum_{t=0}^{\infty} \Psi \nabla_\theta \log \pi_\theta(a|s) \right],$$

where $\Psi$ can be chosen as the advantage function $A^{\pi_\theta}(s, a) = Q^{\pi_\theta}(a, s) - V^{\pi_\theta}(s)$: $Q^{\pi_\theta}(a, s)$ is the action-value function and $V^{\pi_\theta}(s)$ is the state-value function. $V^{\pi_\theta}(s)$ can be estimated by a parameterized model called *critic* (Konda & Tsitsiklis, 2000).

# 4 LIMITLESS ROUTING

In this section, we will describe the way we removed the constraints on the routing network and increase the diversity of the modules, Then, we will describe the MDP formulation and our actor-critic model architecture. Lastly, we will discuss our solution to preventing module collapse.

## 4.1 REMOVING THE CONSTRAINTS OF THE ORIGINAL ROUTING NETWORK

In the original routing network (Rosenbaum et al., 2017), the output size of modules is fixed and each module is required to belong to a particular depth, and thus the total number of modules is fixed at every level of depth of the network. The rationale of a fixed output size of the modules is due to the router takes in the output of a module as part of the input at each timestep. This prevents routing over modules with different output sizes if this constraint is not kept true and thus disallows complex model construction beyond typically a few FC layers with the same fixed output size. The only solution proposed in (Rosenbaum et al., 2017) to circumvent a fixed output size of the modules is by increasing the number of routers per set of modules with the same output size, which is inefficient.

To increase efficiency and allow for more general model construction, we remove the abovementioned constraint by creating an embedding for each module and use the embedding as the part of the input rather than the output of the previous module. More specifically, the input is updated to include the embeddings for the current modules in the model being constructed, as well as the modules that are available to be used in the current timestep in the state. So, the new input to the router is $(v, z, l_0^{model}, \ldots, l_{t-1}^{model}, l_0^{valid}, \ldots l_n^{valid})$, where $v$ is the embedding for the image instance, $z$ is the task embedding, $l_0^{model}, \ldots, l_{t-1}^{model}$ are the embeddings of the modules currently in the model getting constructed, and $l_0^{valid}, \ldots l_n^{valid}$ are the modules that can be used in the current timestep. The modules that can be used in a timestep is based on the current depth of the model getting constructed, the output size of the previous modules (if it is not already used), and the input size of the module.

Due to updating the input of the router, removing the constraint of requiring each module to belong to a particular depth has an easy solution. We simply allow the modules to be used at any depth as long as the output size of the previous module, or the size of the instance if it is the first timestep, matches the input size and it has not already been included in the model. Furthermore, to allow the constructed model itself to be of variable depth size, we include a "PASS" action as it was used in (Rosenbaum et al., 2017), to allow the router to skip adding a module at a given timestep. Note that this design is restricted to be only used on hidden layer modules and it cannot be used on the last timestep if the last module has not been added yet. This is to enforce a valid minimum size modular network, because if we always enable the PASS action, there is a chance that no modules will be selected or incorrect input and/or output sizes that do not match the current task will be picked.

To remove the constraint of requiring the total number of modules to be fixed at every level of depth of the network, we use a conditional transformer model (Keskar et al., 2019; Vaswani et al., 2017) for the router network. This models the conditional probability over the valid modules at a particular depth:

$$P(l_i^{valid} | v, z, l_0^{model}, \ldots, l_{t-1}^{model}), \tag{1}$$

where our control code is given by the task and instance embedding. We sample from this distribution over unused, valid modules while training to encourage exploration of various modular architectures.

## 4.2 MDP FORMULATION

We formulate our problem as a Markov Decision Process (MDP) where the states are defined by the tuple $(v, z, l_0^{model}, \ldots, l_{t-1}^{model})$, the actions are the potential next modules $(l_0^{valid}, \ldots, l_n^{valid})$, the rewards are the negative loss of the constructed model, and the transition probability function is deterministic as it is only based on the routers decisions. The routing network was trained using Proximal Policy Optimization (PPO) (Schulman et al., 2017) using an actor-critic approach with

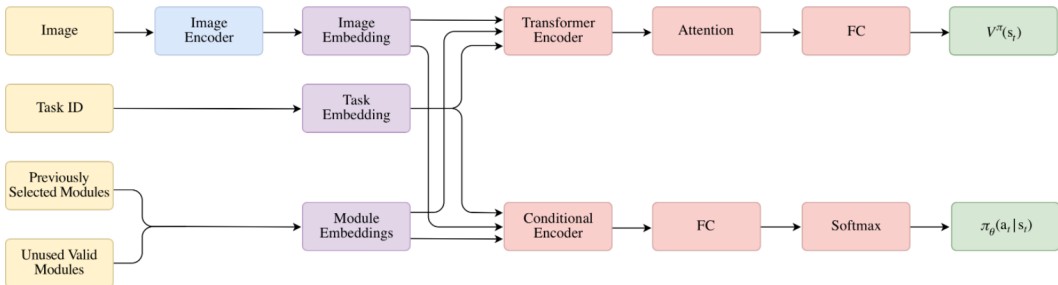

Figure 1: The LRN architecture using actor-critic approach. The state is represented by the image instance, task, and previously selected modules and the actions are over the valid, unused modules.

generalized advantage estimation (GAE) (Schulman et al., 2018) for the advantage estimate. Since the critic has variable input size that depends on the timestep, we make use of a simple attention mechanism (Bahdanau et al., 2016) without the context to combine the embeddings representing the state to acquire a fixed sized state embedding.

Specifically, we first use a transformer encoder (Vaswani et al., 2017) to perform self-attention over the state:

$$\hat{s}_t = \text{Encoder}(s_t). \tag{2}$$

Then, scores are computed over the elements of the state:

$$e_j = W_e \hat{s}_{tj}, \tag{3}$$

where $W_e$ is a vector of weights used to map the embeddings representing the state to a single scalar weight, $e_j$, and $\hat{s}_{tj}$ is of the embeddings for the state (i.e., the instance embedding, task embedding, and the embeddings modules currently in the network).

Next, the scores are normalized into weights using a softmax distribution:

$$\alpha_j = \frac{\exp(e_i)}{\sum_j \exp(e_j)}. \tag{4}$$

Finally, these weights are used to combine the embeddings and output the state-value for the current timestep t:

$$V^{\pi_\theta}(s_t) = W_c \sum_j \alpha_j \hat{s}_{tj}, \tag{5}$$

where $W_c$ is a vector of weights used to map the weighted combination of state embeddings to a state-value prediction. Both $W_c$ and $W_e$ may be represented by a FC layer with a single unit.

The full LRN actor-critic router architecture is given in Figure 1. We used an image encoder that is comprised of a couple convolutional layers followed by a FC layer to convert the instance image into an embedding that has the same size as the task and module embeddings.

## 4.3 PREVENTING MODULE COLLAPSE

Model collapse (Kirsch et al., 2018; Rosenbaum et al., 2019) occurs when the routing network lacks diversity in its selection of modules and collapses to only choosing a fixed subset of modules. There have been many proposed solutions, but all assume the original routing method. Thus, we formulate a novel and simple approach as a reward bonus to prevent module collapse—the reward is given to each module selection proportional to its normalized count over a batch:

$$r_i^{bonus} = 1 - \frac{c_i - c_{min}}{c_{max} - c_{min}}, \tag{6}$$

where $r_i^{bonus}$ is the reward for module $l_i$, $c_i$ is the number of times that $l_i$ has been selected over the entire batch of episodes, and $c_{min}$ and $c_{max}$ are the min and max count over all the modules, respectively. Everytime the module is used in a batch of instances, the corresponding timestep's reward is augmented with the same $r_i^{bonus}$ reward.

The auxiliary reward bonus gives higher reward to modules that are not used frequently over a batch, and a low or near-zero reward to those that are used the most. This encourages the usage of modules that were not used much in a batch and helps prevent over constructing the current best model for each instance. Thus, this reward shaping helps favor diversity of module selection, and therefore helps prevent module collapse. In summary, the final reward for each timestep and example in a batch is as follows:

$$r_{tk} = r_k + r_i^{bonus}, \tag{7}$$

where $r_{tk}$ is the reward for timestep $t$ in example $k$ in the current batch, $r_k$ is the negative loss of the $k^{th}$ constructed model, and $r_i^{bonus}$ is the reward bonus for module $l_i$.

## 5 EXPERIMENTS AND RESULTS

In this section, we will first explain the benchmarks and experiment setup details we used to evaluate our approach. Next, we will describe the modules we used in all the experiments. Then, we will discuss our hyperparameter tuning results for the maximum depth of the constructed model and the number of duplicate modules and results of our solution to preventing module collapse. Lastly, we will compare our results to the original routing network against all the benchmarks.

### 5.1 EXPERIMENT SETUP

We test over the three benchmarks discussed in (Rosenbaum et al., 2017), namely MNIST-MTL, MIN-MTL, and CIFAR-MTL.

- MNIST-MTL uses the MNIST dataset (Lecun et al., 1998), which consists of images of hand-written digits. The task is usually converted to a set of one-vs-all binary classification task for each digit (10 tasks in total). The training set consists of 1 000 positive examples for every task and 1 000 negative examples from every other classes, resulting in a total number of 9 000 negative examples. The test set is created by randomly sampling 200 examples per task from a different set of examples that were not present in the training set.

- MIN-MTL is derived from Mini-ImageNet (Ravi & Larochelle, 2016; Vinyals et al., 2017), which is a subset of ImageNet (Deng et al., 2009). We use the first 50 labels of ImageNet and group them into 10 tasks with 5 labels each. We use 800 examples per task for the training set and 50 per task for the test set.

- CIFAR-MTL uses the CIFAR-100 image classification dataset (Krizhevsky et al., 2009). We create one task for each of the 20 coarse labels and include 500 examples for each of the 5 corresponding fine labels. So, in total, 2 500 examples are created per task for the training set and 500 per task for the test set.

In all the experiments, we train for a total of 100 epochs on the training set with a batch size of 128, a mini-batch size of 64, and 1 mini-batch epoch for PPO. We compute the accuracy over the test set after every epoch. The transformer uses a singular encoder and decoder with 4 heads, and we set $d_{model} = 256$ and $d_{dff} = 512$ as in the transformer model (Vaswani et al., 2017). All the embeddings and hidden units of FC layers are of size 256. We set $\gamma = 0.99$ and $\lambda_{GAE} = 0.95$ for the return and advantage computation. We set the initial learning rate for both training the modules and router to 0.0001 and anneal them by a factor of 0.97 after every epoch.

## 5.2 MODULE DESIGN

As LRN removes the constraints of the original routing network, our router can perform routing on every layer of the neural network. Thus, we came up with a set of modules that can be used to construct a fully routed modular network. There are mainly three different types of modules: those that can be used in the first layer, hidden layers, and the output layer. We assume every module is made up of a single FC or convolutional layer. Furthermore, we come up with two hyperparameters that control the maximum depth of the constructed network and the number of duplicates allowed for each module described below.

The input module is a convolutional layer with input channel size matched to the image size. Every convolutional layer, at every depth, contains 32 3x3 filters with stride and padding set to 1. This is to keep the input size equal to the output size to constrain the number of possible models. However, this is not required and can be of any variable size, as long as the design is able to be placed in the constructed network (i.e., its input size matches the output of another module and vice versa). If not, the module could never be used.

The hidden layer modules consist of a convolutional layer and 2 FC layers with 256 hidden units. The first FC layer has input size equal to the output size of the convolutional layers, and the second with input size 256 to allow being connected to the other hidden FC layers.

The output layer modules, which are used as the head of the network, are 2 FC layers. The are identical to the 2 FC layers used as hidden layers, as to allow being connected to either a convolutional or FC layer, but with output size equal to the number of classes.

We apply layer normalization (Ba et al., 2016) after every FC module, besides the output module, and instance normalization (Ulyanov et al., 2017) for convolutional modules. Furthermore, they are both followed by ReLU activation (Nair & Hinton, 2010).

## 5.3 MAXIMUM DEPTH AND DUPLICATES ANALYSIS

As we removed the constraint on the number of modules that can be used and on what layers routing can be performed, we decide to explore what the best maximum depth of the constructed network and the number of duplicates to include for each module discussed previously. We experiment over various combinations of module depth and duplicates for all the benchmarks. The results are shown in Figures 2, 3, and 4, where we use the naming scheme "dup_x_depth_y" to denote x duplicates and y max depth.

The figures on the left give the accuracy over the test set after each epoch, and on the right the number of unused modules. If we jointly increase the maximum depth and the number of modules, the performance will improve, with the best appearing at 10 duplicates per module and the maximum depth of 6. However, increasing the number of duplicates generally slows down the convergence speed. This is because with more modules, the state and action space increases, causing the task complexity to increase as well. Also, if a module is not used in a batch of example, its parameters will not be updated after that batch.

## 5.4 PREVENTING MODULE COLLAPSE ANALYSIS

For our method to prevent module collapse, we compare it to the case when the reward is not used. We test on the "dup_5_depth_4" model as to keep the space of possible modular networks small, but still show the effectiveness of the auxiliary reward bonus. The results are given in Figures 5, 6, and 7 where we append "_no_bonus" to the name of the ones that are not using module collapse prevention.

In the MNIST-MTL and CIFAR-MTL benchmarks, the results show clear evidence of module collapse as the number of unused modules never decreases when the bonus is not used. This also results in the test accuracy increasing by 16% and 28% for MIN-MTL and CIFAR-MTL, respectively. Also, in all the benchmarks, when the bonus reward is used, LRN uses more modules over the test set, thus succeeding in usage of modular diversity.

Although the results are close, MNIST-MTL performs slighter better (about 1%) when the reward bonus for preventing module collapse is not used. We believe this is because MNIST-MTL repre-

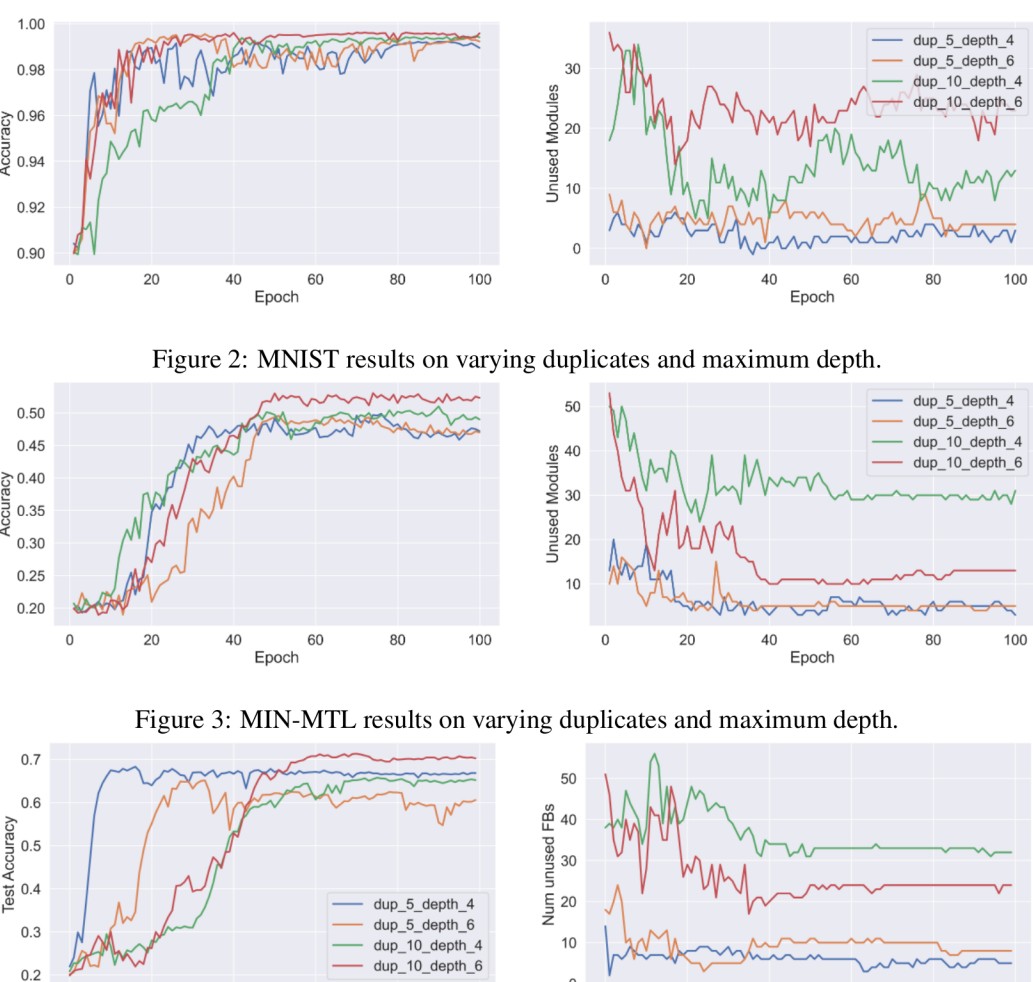

Figure 2: MNIST results on varying duplicates and maximum depth.

Figure 3: MIN-MTL results on varying duplicates and maximum depth.

Figure 4: CIFAR results on varying duplicates and maximum depth.

sents an easier problem and all sub-tasks of MNIST-MTL are very similar, so the positive transfer between tasks is high, while catastrophic forgetting and negative transfer is low. Therefore, training the routing network tends to be easier. Regardless, the results clearly show that the auxiliary reward bonus help prevent module collapse.

## 5.5 COMPARISON TO THE ORIGINAL ROUTING NETWORK

We follow the same procedures as discussed in (Rosenbaum et al., 2017) to recreate the experiment results on the routing network. We use LRN with 10 duplicate modules and a maximum depth of 6 as we found that resulted in the best performance on all the benchmarks. The results against the original routing network is given in Table 1. Our approach outperforms their approach measured over all the benchmarks. Note that we fully route every layer in the modular network, while (Rosenbaum et al., 2017) only routes the last 3 layers while keeping all the initial convolutional layers fixed. So, this shows support for fully routing the network. Furthermore, we use function approximation as opposed to a tabular representation for the router policy. In (Rosenbaum et al., 2017), function approximation has an inferior performance. However, our results show that we rectify this issue by utilizing instance level information—which can only be used by function approximation (e.g., it is not a trivial solution to build a tabular representation for pixel-based data). As function approximation is much more versatile than tabular representation, we expect our approach to have a broad impact on multi-task learning.

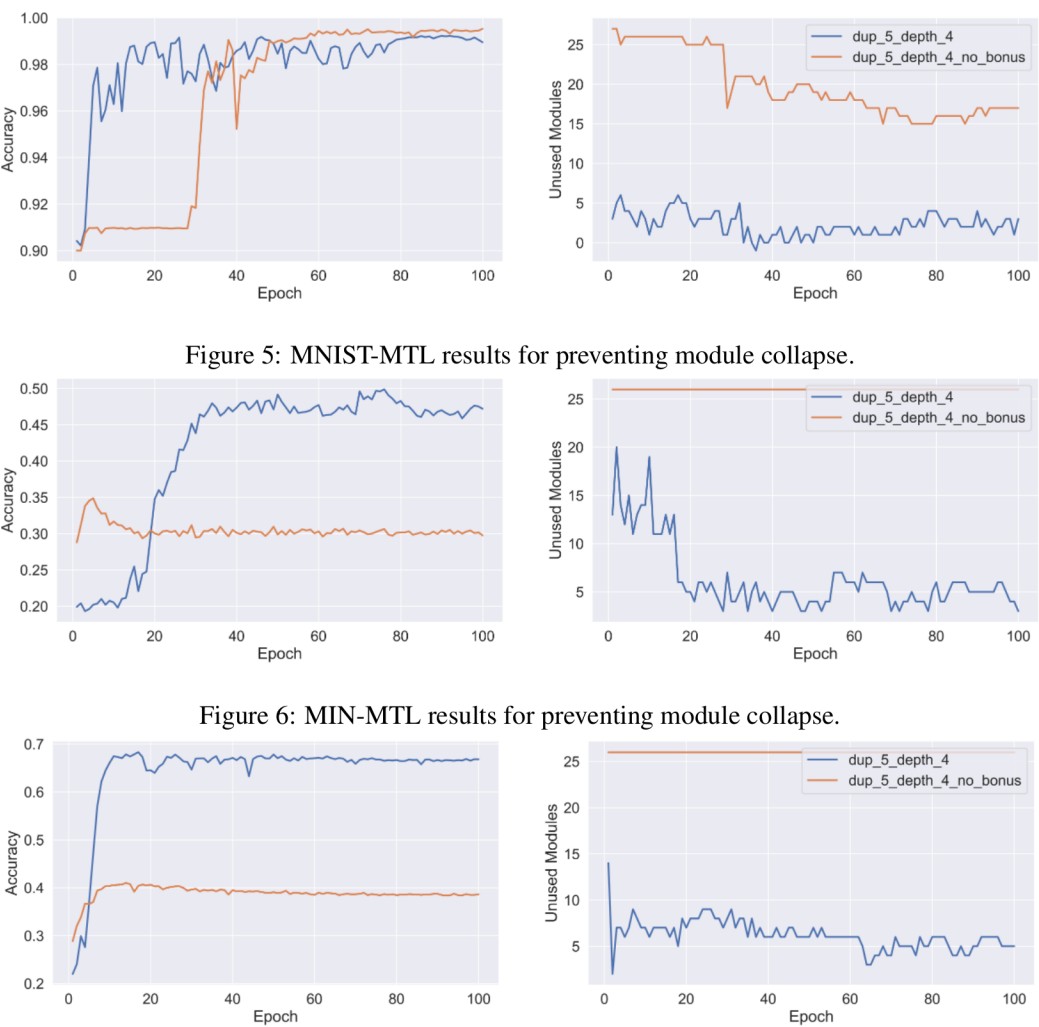

Figure 5: MNIST-MTL results for preventing module collapse.

Figure 6: MIN-MTL results for preventing module collapse.

Figure 7: CIFAR-MTL results for preventing module collapse.

Table 1: Performance comparison to the original routing network. As our experiments suggest, LRN outperforms the original routing network on all the benchmarks.

| | Test Accuracy | | |
|---|---|---|---|
| **Method** | MNIST-MTL | CIFAR-MTL | MIN-MTL |
| Routing Network | 99.0% | 60.0% | 35.9% |
| Limitless Routing Network (ours) | **99.5%** | **70.0%** | **53.0%** |

## 6 CONCLUSION

In this work, we propose Limitless Routing Network (LRN) for multi-task learning. LRN removes the constraints required by the original routing network (Rosenbaum et al., 2017) in terms of fixed output size of modules, and the requirement of each module belonging to a particular depth in the network. As a result, LRN allows more complex model construction and outperforms the original routing network on all the benchmarks. In addition, we provide a simple reward bonus to help circumvent module collapse. We also allow the use of function approximation (instead of tabular representation) in the routing network, which greatly increases the potential of the routing network on complex tasks. We have conducted extensive experiments and analysis of LRN hyperparameters

to demonstrate the effectiveness of our approach. In the future, we would like to provide more in-depth experimentation on various multi-task learning domains to further show the generality of our approach. Furthermore, we intend to extend the framework to use more complex modules that can each be composed of multiple layer or provide a unique forms of perception (e.g., using complex modules made up of convolutional neural networks for visual-related tasks or transformers for language-related tasks). Lastly, we would also like to apply LRN to multi-modal multi-learning tasks where LRN can have an array of modules for each modality.

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
