# OpenReview forum: "LRN: Limitless Routing Networks for Effective Multi-task Learning"
_ICLR.cc/2022/Conference — ICLR 2022 Submitted_

### Official Review · Reviewer_ewVv · 2021-10-27

**Correctness:** 3
**Technical Novelty And Significance:** 2
**Empirical Novelty And Significance:** 2
**Recommendation:** 3
**Confidence:** 4

**Main Review:**

This work improves the existing routing network with several constraints by proposing a limitless routing network that removes the constraints using an actor-critic approach. However, it is incremental because the main structure is based on the original routing network, and on top of it, they add more embeddings not to use the previous module with a fixed output size. From this, I think the contributions of this work are rather limited. The additional suggestion to prevent module collapse is straightforward. Rigorous analyses with other works sharing a similar idea/motivation are missing as well.

The writing in Section 4 is rather hard to follow as it does not describe the meaning of variables and indices when they are first introduced, for example, l_0^model, 0_l^valid, t, and n. I could roughly understand what they are by looking at Figure 1.
What does \hat{s_{tj}} in Eq. (3) mean and how it is different from \hat{s_t} in Eq. (2)?
I feel there is a gap between the description and the figure for the proposed method. Descriptions on how the transformer encoder and attention are used and when we use the conditional encoder are missing in the main text.
Before Eq. (7), what does ‘example in a batch’ mean in the context? In Eq. (7), why do you put only the reward bonus for the module l_i to the k-th model?
Since there are many unclear expressions, the authors should improve the paper.

The presented experiments are not effective because this paper contains two analyses for the proposed method itself and one comparison to the baseline without other approaches. In addition, the experiments are all on small-scale datasets and do not contain many tasks. From those, it is hard to judge whether the proposal is promising or not. More thorough experiments should be conducted to make the paper stronger.


**Summary Of The Paper:**

This paper presents an improved routing network for multi-task learning by removing constraints that are applied in the existing routing network. This is achieved by introducing an actor-critic approach equipped with two different encoders. It also presents a simple approach to avoid module collapse based on a reward for each module. Experimental results show that the proposed approach performs better than its baseline approach on several MTL benchmark datasets.

**Summary Of The Review:**

I have some concerns on this paper; the novelty of the proposed method and a limited set of experiments. The authors should address the concerns clearly and compare the proposal with other similar works.

---

### Official Review · Reviewer_v8R3 · 2021-11-02

**Correctness:** 2
**Technical Novelty And Significance:** 2
**Empirical Novelty And Significance:** 1
**Recommendation:** 3
**Confidence:** 5

**Main Review:**

The paper is well written and easy to follow. They propose to route the instance among modules. And modules are not limited to a certain depth and can be connected in different orders, which is called as limitless routing network. The idea is novel and interesting. Their use of actor-critic RL algorithm to optimize the router is also a solid choice. They also introduce a reward to prevent the mode collapse.

However, the experiments are not adequate to substantiate their claim. First, the datasets used in the paper are pretty simple, e.g. CIFAR and MNIST. Second, they do not compare to many SOTA MTL methods, such as residual adapter [A], cross-stitch [B], ASTMT [C], MTAN [D] and etc to show their performance. Also, they miss the discussion of computation cost and memory efficiency in the experiment, which are important metrics to evaluate the MTL algorithms.

[A] Learning multiple visual domains with residual adapters
[B] Cross-stitch Networks for Multi-task Learning
[C] Attentive Single-Tasking of Multiple Tasks
[D] End-to-End Multi-Task Learning with Attention


**Summary Of The Paper:**

The paper proposes a limitless routing network to route the instance in a certain task through a couple of modules in the multi-task learning scheme. They adopt a policy network to generate the probability of all available modules for the next step based on instance embedding. task id and previous modules. They optimize the policy network with the actor-critic RL algorithm. They evaluate their approach on MNIST-MTL, Min-MTL and CIFAR-MTL.

**Summary Of The Review:**

The proposed method looks novel and reasonable but the experiments cannot well support the effectiveness of the proposed method.

---

### Official Review · Reviewer_CZQn · 2021-11-02

**Correctness:** 4
**Technical Novelty And Significance:** 1
**Empirical Novelty And Significance:** 1
**Recommendation:** 3
**Confidence:** 4

**Main Review:**

Although the paper identifies 3 constraints with the original routing network, it is not clear why these constraints are limiting. For example, one of the constraints is that modules could only be used at a specific depth. However, this seems to be a reasonable assumption. Why would a network module that is trained to process low-level features be used deeper in the inference path (ie to process higher-level features)? The proposed approach does not enforce this explicitly, but the authors could add an analysis to show what is the preferential position (in the inference graph) for each block.

The results for routing networks for MIN-MTL in table 1 (35.9%) seems to be inconsistent with the original paper (which presents an accuracy of ~57% - original paper figure 7.) What is the justification for this difference?

This paper does not compare with any other models, other than routing networks. Multi-task learning is a widely studied field, and thus many different approaches have been proposed to address this problem. A more extensive comparison to prior work is warranted.

**Summary Of The Paper:**

The paper proposes an improvement to routing networks (Rosenbaum 2017): a multi-task learning model that builds the neural network architecture from set of modules, which can be adaptively chosen conditioned on the input sample and target task. This paper identifies several constraints with the original routing networks, namely 1) each module output dimension was constant, 2) modules were depth specific, and 3) the total number of modules available per level was fixed. The paper then proposes a model that lifts these constraints. It also introduces a bonus reward to encourage all modules to be equally used.

**Summary Of The Review:**

The paper presents an improvement to a previous multi-task learning model called routing networks. The proposed improvements are not well motivated. The experimental validation is also limited given the lack of comparisons to prior work. Thus due to the limited novelty and poor experimental validation, I leaning towards rejection.

---

### Official Review · Reviewer_Xh12 · 2021-11-03

**Correctness:** 3
**Technical Novelty And Significance:** 2
**Empirical Novelty And Significance:** 2
**Recommendation:** 3
**Confidence:** 4

**Main Review:**

Strength:
1. This paper introduced a novel idea of using a transformer in the router RL for MTL which solves the input size limitation of the traditional routing network.
2. The routing module is agnostic to the input type so it can be applied to different types of networks. This can enlighten the application of transformers in RL.

Weakness
1. The contributions of this paper is sort of limited since the routing network is not the SOTA architecture for MTL. The routing network is designed for limited classification problems and is hard to be scaled to the advanced MTL problems including detection, segmentation, depth estimation, etc.
2. The paper is not well written. The paper does not introduce the challenges of the routing network clearly. Given this paper is to improve the baseline routing network, it is necessary to revisit the design and challenges of the routing network in detail.
3. Some parts of the paper are not clearly presented. For example, the conditional encoder in Figure 1 is not introduced. How to apply the routing module to all the layers is not introduced in the implementation.
4. Although the proposed approach outperforms the routing network baseline, the paper lacks the performance comparison with the SOTA MTL work. It is hard to verify the value of using the proposed algorithm.
5. The paper missed the model’s effectiveness evaluation of the proposed approach. Applying the transformer based routing module can make the model size very big. The model can suffer from overfitting issues which need further ablation studies.

**Summary Of The Paper:**

This paper proposed a transformer based routing module to improve the routing network for multitask learning. The main contributions of this paper can be summarized as follows.
1. The paper introduced transformer based routing to solve the input length limitation in the previous routing network.
2. The transformer based routing network can be generally applied to different types of layers.
3. This paper resolved the module collapse by only route to a subset of modules regardless of the input.

**Summary Of The Review:**

This paper proposed a novel idea for using the transformer in the routing network. However, the paper missed many important evaluations including comparing it with SOTA MTL approaches, model effectiveness, etc. Besides, the paper was not well presented which makes it hard to follow. Overall, the paper needs a major improvement.

---

### Decision · Program_Chairs · 2022-01-20

**Decision:**

Reject

**Comment:**

This paper proposed a transformer based routing network which removes the constraints in the original routing network such as the depth of a network. Multi-Task learning (MTL) based on routing has been an interesting topic in the deep learning research community.  Our reviewers have serious concerns on the experiments. The presented empirical results do not seem to be able to sufficiently support the claims in this paper. Comparing with SOTA MTL methods is needed to make the proposed method convincing.